# Renal Consequences of Gestational Diabetes Mellitus in Term Neonates: A Multidisciplinary Approach to the DOHaD Perspective in the Prevention and Early Recognition of Neonates of GDM Mothers at Risk of Hypertension and Chronic Renal Diseases in Later Life

**DOI:** 10.3390/jcm8040429

**Published:** 2019-03-28

**Authors:** Maria Cristina Aisa, Benito Cappuccini, Antonella Barbati, Graziano Clerici, Elisabetta Torlone, Sandro Gerli, Gian Carlo Di Renzo

**Affiliations:** 1Department of Surgical and Biomedical Sciences, Section of Obstetrics and Gynecology, University of Perugia, 06132 Perugia, Italy; antonella.barbati@unipg.it (A.B.); sandro.gerli@unipg.it (S.G.); giancarlo.direnzo@unipg.it (G.C.D.R.); 2GeBiSa, Research Foundation, 06132 Perugia, Italy; benitocappuccini@hotmail.com (B.C.); g.clerici@cda-adc.it (G.C.); elisabetta.torlone@ospedale.perugia.it (E.T.); 3Second Department of Obstetrics and Gynecology, University of Sechenov, 119992 Moscow, Russia; 4Department of Internal Medicine and Endocrine and Metabolic Sciences, University of Perugia, 06132 Perugia, Italy; 5Centre of Perinatal Medicine, University of Perugia, 06132 Perugia, Italy

**Keywords:** gestational diabetes, total renal volume, cortical volume, *N*-acetyl-β-d-glucosaminidase, cathepsin B, maternal weight gain

## Abstract

Fetal exposure to gestational diabetes mellitus (GDM) seems to stimulate a negative impact on the kidneys. Renal volumes and urinary biomarkers of renal function and tubular impairment and injury were evaluated in 30–40-day old newborns of GDM mothers (*n* = 139) who needed insulin therapy during pregnancy. We found that neonates of mothers who maintained strict control over normoglycemia (*n* = 65) during pregnancy and fulfilled the other criteria of the GDM management program showed no differences compared to control (*n* = 55). Conversely, those (*n* = 74), whose mothers did not maintain glycemic control and were not compliant to the management program, exhibited significantly lower levels of renal volumes and higher activity of *N*-acetyl-β-d-glucosaminidase and cathepsin B. Differences due to maternal pre-gestational and gestational body mass index (BMI) as well as to maternal weight gain were demonstrated. Our findings indicate that a multidisciplinary approach, which involves an appropriate management of GDM, prevents the negative effects of GDM on the kidneys at 30–40 days of postnatal age, indicating the fundamental role of glycemic control, as well as of an adequate range of maternal weight gain. Total renal volume, cortical volume, and urinary activity of *N*-acetyl-β-d-glucosaminidase and cathepsin B may be suggested as indicators for the early recognition of GDM neonates at long-term risk of hypertension and kidney disease.

## 1. Introduction

It is now well established that conditions during fetal or early postnatal development influence the individual’s risk of developing non-communicable diseases in later life (Developmental Origins Of Health and Disease (DOHaD) paradigm) [1,2]. As a consequence, interventions to optimize maternal, fetal, and child health are extremely important in order to prevent adult non-communicable diseases.

More recently, a significant impact on global morbidity and mortality due to hypertension and chronic kidney disease [3,4] has emerged and has been correlated with adverse events experienced in utero that can affect fetal kidney development and reduce nephrogenesis [5,6,7,8]. Low nephron endowment has been proposed as a determinant agent of these diseases, as it may generate a vicious cycle of progressive loss of functioning units [9,10,11] or constitute a “factor of vulnerability” to additional insults during fetal, perinatal, and neonatal life [6], causing a major risk of renal function impairment, long term renal diseases, or high blood pressure [5,6,7,8,9,10,11].

In experimental models, maternal hyperglycemia is associated with reduced nephron number, raised blood pressure, microalbuminuria, and diminished glomerular filtration rate in offspring [12]. In adult children whose mothers had diabetes, compared with those who had a diabetic father, renal functional reserve was decreased, suggesting a reduction in nephron number that was acquired during exposure to gestational diabetes [13]. Maternal diabetes is also associated with a threefold increased risk of renal agenesis and dysgenesis [14]. Furthermore, gestational diabetes is sometimes associated with high birth weight in infants, which is a known risk factor for subsequent hypertension, type 2 diabetes, renal disease, and cardiovascular disease, although the effect on nephron number is unknown [15]. Additionally, a direct correlation between reduced nephrogenesis, proteinuria, and gestational diabetes mellitus (GDM) in 3-year-olds has been recognized as a cause of kidney injury in offspring [16], and remarkably, a significant association between GDM and the rate of cardiovascular hospitalizations, including hypertensive disorders, in the offspring has recently been demonstrated in a population-based cohort study with up to 18 years of follow up [17].

GDM has recently reached epidemic proportions worldwide and dysregulation of glucose metabolism is found in up to 15% of pregnancies. Accordingly, with such impressive data and in agreement with the DOHaD concept, the recent guidelines support a recommended GDM management, which involves a multidisciplinary approach to achieve a healthy childhood, adolescence, and future life [18,19,20].

To date, no data concerning renal development and function in the early phase of postnatal period in newborns of GDM mothers have been reported. In this study we evaluated possible negative consequences of GDM on renal adaptation in term infants at 30–40 days of age. We also examined potential differential effects associated with different management approaches of GDM during pregnancy and the impact of maternal BMI in this population.

We considered renal development and function as well as tubular impairment and injury or dysfunction. Renal development was assessed by measuring total renal and cortical volumes, which are the primary surrogate markers of nephron number [9,10,11,16,21,22]. Renal physiology and possible impairment and injury were evaluated by determining urinary parameters of glomerular and tubular function, as well as of tubular impairment and injury or dysfunction. These included urinary level of albumin, β2-microglubulin, and the activity of *N*-acetyl-β-d-glucosaminidase and cathepsin B.

Urinary albumin is a well-known marker of glomerular permeability [23,24], also representing a powerful predictor of kidney disease [25,26]. The β2-microglobulin is believed to reflect renal proximal tubular function in neonates, and in diabetic conditions, increases in urine [26,27,28]. Similarly, higher levels of *N*-acetyl-β-d-glucosaminidase and cathepsin B have been seen following tubular damage or dysfunction [29,30,31]. Additionally, urinary *N*-acetyl-β-d-glucosaminidase and cathepsin B have been reported to be significantly enhanced in premature and intrauterine growth restriction (IUGR) neonates at 30–40 days of corrected age, and significantly and negatively correlated with renal volume and cortical volume [32].

The urinary activity of β-glucuronidase and legumain were also considered in order to establish a possible general effect of GDM on tubular lysosomal enzyme excretion or perturbation on tubular maturation, respectively. Levels of urinary β-glucuronidase and *N*-acetyl-β-d-glucosaminidase have been seen to increase and associate in some diseases of the urogenital tract [33], whereas legumain plays an important role in the function of renal proximal tubular cells, such as the absorption of macromolecules and the remodeling of extracellular matrix proteins [34,35].

## 2. Experimental Section

### 2.1. Study Design

The characteristics of the present study were resumed in Table 1.

### 2.2. Neonates

A group of 194 newborns at 30–40 days postnatal period, born at term, were examined. Of them, 139 were of GDM mothers and 55 were healthy of healthy mothers and used as matched controls. The neonates were highly selected in order to exclude all those at risks of renal defects at birth. Thus, the exclusion criteria were: prematurity, IUGR, twins, asphyxia, sepsis, macrosomia, any neonatal malformation including those of kidney, acute kidney injury (AKI), maternal accelerated weight gain in the first trimester, maternal hypertensive disorders, preeclampsia, maternal smoking, maternal alcohol use, maternal caffeine abuse, pre-existent renal diseases in both parents and in family history, maternal diabetes mellitus type I and type II. To avoid influence by birth weight, placenta weight, and maternal age on total renal mass and function [36], we selected neonates with similar values of these variables (Table 2). The population showed an Apgar score value ≥7 and ≤10, at the 1st and 5th minute.

After the enrolment, the GDM neonates were divided into subgroups and classes according to the categorization of the corresponding mothers (Section 2.3 and Figure 1).

### 2.3. GDM Mothers

We enrolled neonates of GDM mothers (*n* = 139), in whom insulin therapy was necessary, excluding those affected by diabetes mellitus type I and type II. The GDM mothers attended the center that specialized in the care of pregnant women with diabetes at the Santa Maria della Misericordia Hospital, in Perugia, Italy.

Diagnostic evaluation and management followed the guidelines of Italian Diabetologist Association and Italian Society of Diabetology [20]. GDM was diagnosed by 75 g Oral Glucose Tolerance Test and insulin treatment was indicated when Fasting Plasma Glucose (FPG) was higher than 5.1 mM (92 mg/dL) and/or 2 h Postprandial Glucose (PPG) was higher than 7.2 mM (130 mg/dL). Patients were managed using a multidisciplinary team approach [20]. The main goal of the treatment was to maintain blood glucose as near to normal as possible. The recommended glycemic targets were: FPG and 1 h PPG less than 4.9 mM (90 mg/dL) and 7.2 mM (130 mg/dL), respectively [20]. The management program aimed at ensuring an adequate maternal weight gain and fetal growth, optimizing glycemic control, avoiding ketoacidosis, and reducing glucose levels after meals. The GDM patients were followed-up by a team and were included into an educational program in order to customize weight gain and calorie intake, and establish their needs in terms of type and distribution of carbohydrates, optimal protein, fat and micronutrient intake, and amount and type of physical activity. They were taught by nurses how to check their own blood glucose levels and were monitored by a specialist at a diabetes outpatient clinic one week after the diagnosis and every 2–3 weeks following that.

All GDM mothers exhibited values of the glycosylated hemoglobin (HbA1C) ≤6%.

According to the compliance or noncompliance with the guidelines of the management program [20], the GDM mother group was distinguished into the subgroups of Compliant (*n* = 65) and Noncompliant (*n* = 74) GDM mothers, respectively. In detail, the main objective criterion to define subjects as Compliant or Noncompliant was the glycemic control, with reference to the recommended targets. Thus, the Compliant subgroup included pregnant mothers who adhered to the nutritional and therapeutic indications and showed mean glycemia values ranging under the recommended targets. The Noncompliant subgroup, in contrast, included subjects who did not reach the glycemic targets, and either followed or did not follow the dietary indications, and/or ensured or did not ensure the appropriate weight gain.

In all GDM patients, maternal pre-gestational and gestational BMI, as well as gestational weight gain, were recorded. Pre-gestational and gestational BMI were defined as weight before conception or during pregnancy in kilograms divided by height in meters squared (kg/m^2^). The BMI classification was based on the WHO cut-off points (underweight <18.5 kg/m^2^, normal weight from 18.5 to 24.9, overweight from 25 to 29.9 and obese >30 kg/m^2^). Gestational weight gain (kg) was defined as the subtraction between the actual weight at delivery and the initial weight just before becoming pregnant.

During the course of the present study, according to the pre-gestational and gestational BMI, the subgroups of Compliant and Noncompliant GDM mothers were divided in the following classes:(a)Class 1: including mothers characterised by both pre-gestational and gestational BMI < 30;(b)Class 2: including mothers characterised by pre-gestational BMI < 30 and gestational BMI > 30;(c)Class 3: including mothers characterised by both pre-gestational and gestational BMI > 30.

Values of BMIs are reported in Table 3.

Institutional review board approval was obtained for data collection and mothers were informed and gave specific consent to the study.

### 2.4. Renal Mass Parameters

Total renal volume and cortical volume were reconstructed and estimated by echo 3-D combined with Virtual Organ Computer-Aided Analysis software (VOCAL) (Vocal II, GE ULTRASOUNDS, USA), a technology that has been shown to be highly reproducible and accurate for the assessment of organ volumes in fetal life and throughout childhood [37]. Measurements were obtained as an average of four repeated estimations by a blinded sonographer with intra- and inter-operator variability less than 5%.

### 2.5. Urinary Biomarkers

For each child, a first morning urine sample was obtained (using a U-bag collection device) and immediately stored in ice to avoid denaturation. Once transferred to our laboratory, measurement of leukocytes and nitrite were tested with a multiple test strip (Combi-Screen PLUS, Analyticon Biotechnologies AG, Lichtenfels, Germany) to exclude possible urinary concomitant infections. Samples were then centrifuged at 5000 rpm for 20 min at 4 °C before storage at −80 °C for later analysis.

All biochemical parameters under investigation were expressed as ratio to urinary creatinine in order to avoid differences in urinary flow rate.

Urinary creatinine was measured using an enzymatic method (Advia ECREA_2, 04992596, performed on Advia 1800 analyzer Siemens) and expressed as mmol/mL.

Microalbumin (mg/mL) and β2-microglobulin (μg/mL) were determined by an immunonephelometric method (BN II Siemens, using human albumin or β2-microglobulin as standard). Data were expressed as creatinine ratio (mg of microalbumin or μg of β2-microglobulin/mmol creatinine).

*N*-acetyl-β-glucosaminidase, cathepsin B, β-glucuronidase, and legumain activities were detected using the specific fluorescent substrates, as previously described [35,38,39,40], i.e., 4-methylumbelliferyl-2-acetamido-2-deoxy-β-d-glucopyranoside (Sigma-Aldrich, Saint Louis, MO, USA) 1 mM in 0.1 mol/L citrate/0.2 mol/L phosphate buffer pH 4.5 for *N*-acetyl-β-glucosaminidas; Z-Arg-Arg-NH-MEC (Bachem, Switzerland) 12 μg/mL in 0.1 M Na-phosphate buffer pH 6.3, 1 mM EDTA, 0.1 mM DTT for cathepsin B; 4-methylumbelliferyl-b-d-glucuronide (Sigma-Aldrich, Saint Louis, MO, USA) 3 mM in 0.1 mol/L citrate/0.2 mol/L phosphate buffer pH 4.5 for β-glucuronidase; Z-Ala-Ala-Asn-MEC (Bachem, Switzerland) 10 μM in 50 mM MES pH 5.0, 125 mM NaCl, 1 mM EDTA, 1 mM DTT, for legumain. For the assays, urine was appropriately diluted to avoid possible interference with inhibitors and incubated at 37 °C with the substrate solutions. The reactions were stopped by adding the specific stopping solutions (i.e., 0.2 M glycine-NaOH buffer, pH 10.4, in the case of *N*-acetyl-β-glucosaminidase and β-glucuronidase, or 0.1 M monoiodoacetic acid in 1 M Tris–HCl buffer pH 8.0, in the case of cathepsin B and legumain). Fluorescence of the liberated 4-methylumbelliferone or 7-amino-4-methylcoumarin was measured on a Perkin-Elmer LS3 fluorimeter, with excitation at 360 nm and emission at 446 nm for *N*-acetyl-β-glucosaminidase activity and β-glucuronidase, or 370 nm and 460 nm for cathepsin B and legumain. The fluorimeter was calibrated using 4-methylumbelliferone or 7-amino-4-methylcoumarin solution in 0.2 M glycine buffer (pH 10.4) or 0.1 M monoiodoacetic acid in 1 M Tris–HCl buffer (pH 8.0), respectively. The activities were corrected for urine creatinine concentration and then expressed as International Units (IU)/min mmol creatinine in the case of *N*-acetyl-β-glucosaminidase and β-glucuronidase, or IU/h mmol creatinine in the case of cathepsin B and legumain. One IU of activity is the amount of enzyme that hydrolyzes 1 μmol of substrate at 37 °C.

### 2.6. Statistical Analysis

Data analysis was carried out and graphs were drawn using GraphPad Prism version 6.01 statistical software. The D’Agostino-Pearson normality test was used to assess the normal distribution of variables. As variables were found not normally distributed, comparison between two groups was performed using the non-parametric Mann-Whitney test and multiple comparisons between more than two groups were performed using non-parametric Kruskal-Wallis one-way ANOVA with Dunn’s ad hoc posttest. Possible predictive accuracy of variables was quantified as the area (AUC) under the receiver operating characteristics (ROC) curve. ROC curves were constructed considering values of control population versus values of Noncompliant GDM mother neonates.

## 3. Results

### 3.1. Characteristics of the Study Population

Some variables that could influence renal mass parameters and function in the study population [36] are detailed in Table 2. Results were comparable.

### 3.2. Renal Mass Parameters and Urinary Biomarkers in Neonates of GDM Group, Compliant, and Noncompliant GDM Mother Subgroups and Control

Statistical data and results of comparison analysis of the variables investigated are reported in Table 4.

Comparing GDM and control group, GDM neonates showed a significant reduction of both total renal and cortical volumes (Table 4a) and a significant increase of *N*-acetyl-β-d-glucosaminidase and cathepsin B activities (Table 4b), whereas levels of albumin and β2-microglobulin were unchanged (Table 4b).

Multiple comparison analysis between control group and the subgroups of Compliant (*n* = 65) and Noncompliant GDM mother neonates (*n* = 74) showed that total renal volume and cortical volume of Noncompliant GDM mother neonates were significantly lower than control, and the subgroup of Compliant GDM mothers (Table 4a). No differences were seen between control and Compliant GDM mother newborns (Table 4a). Concerning renal biomarkers, *N*-acetyl-β-d-glucosaminidase and cathepsin B activity exhibited significantly higher levels in the Noncompliant GDM mother neonates versus control (Table 4b), whereas they were unchanged in the control group and the subgroup of Compliant GDM mother neonates (Table 4b). Urinary albumin and β2microglobulin were similar in all cases (Table 4b).

### 3.3. Evaluation of Urinary β-Glucuronidase and Legumain Activities in the Subgroups of Compliant and Noncompliant GDM Mother Neonates and Control

As the activity of the lysosomal enzymes *N*-acetyl-β-d-glucosaminidase and cathepsin B were significantly increased in the subgroup of Noncompliant GDM mother neonates, to establish a possible general effect of GDM on tubular lysosomal enzyme excretion or on perturbation on tubular maturation, we assayed the urinary activity of β-glucuronidase and legumain in the two subgroups of GDM neonates and control. We found that levels of these activities were unchanged in all neonates examined (Table 5).

### 3.4. Recognition of Three Classes in Each Subgroup of GDM Neonates Based on the Presence or Absence of Maternal Pre-gestational and Gestational Obesity and Multiple Comparison of the Parameters Investigated

Pre-gestational and gestational obesity (BMI > 30) have been seen to induce negative effects on neonates [41,42,43,44,45,46,47]. To evaluate possible and differential effects of maternal pre-gestational and gestational obesity (BMI > 30) concurrent with GDM on renal mass parameters and urinary *N*-acetyl-β-d-glucosaminidase and cathepsin B activity, we recognized three classes for both subgroups of GDM neonates, according to those of the corresponding mothers (Section 2.3).

Analysis of renal mass parameters demonstrated that, compared to control, total renal volume and cortical volume were unchanged in Classes 1–3 of the subgroup of Compliant GDM mother neonates (Figure 2). In the Noncompliant GDM mother one, there was a different trend. In neonates of Classes 1 and 3, total renal volume was significantly decreased compared to control and Class 2. Cortical volume showed a similar tendency, however, the difference between Class 1 and Class 2 was not statistically significant (Figure 2).

For *N*-acetyl-β-d-glucosaminidase and cathepsin B activity, multi comparison analysis indicated no differences in control and in the three classes of the Compliant GDM mother neonate subgroup (Figure 3). On the other hand, in the Noncompliant GDM mother neonate subgroup, they exhibited significantly higher activity in Classes 1 and 3 compared to control (Figure 3). *N*-acetyl-β-d-glucosaminadase activity, in addition, was significantly augmented in Class 3 with respect to Class 2 (Figure 3).

### 3.5. Maternal Weight Gain in Classes 1–3 of Both Subgroups of GDM Neonates

As we found that obesity did not influence renal mass parameters and *N*-acetyl-β-d-glucosaminadase and Cathepsin B activity in the classes of Compliant GDM mother neonates, and, mostly, in Class 2 compared to Classes 1 and 3 of the Noncompliant GDM mother neonate subgroup, we then investigated if these trends were related to maternal weight gain, as this parameter influences fetal health [44,45,46,47].

Data are illustrated in Figure 4. In brief, in the subgroup of Compliant GDM mother neonates, no significant differences occurred among all three classes, while they did occur in the Noncompliant GDM mother ones. In this case, Classes 1 and 3 presented statistically lower maternal weight gain compared to Class 2. In addition, comparing the corresponding classes of the two subgroups, Classes 1 and 3 of Noncompliant GDM mother neonate subgroup showed significantly lower values than corresponding Classes 1 and 3 of the other subgroup, whereas Class 2 was similar for both (Figure 4).

### 3.6. ROC Curve Analysis of Renal Volume, Cortical Volume, *N*-acetyl-β-Glucosaminidase, and Cathepsin B Activity

Possible diagnostic efficiency of renal volume, cortical volume, *N*-acetyl-β-glucosaminidase, and cathepsin B activity as risk factors for renal disease in later life of GDM neonates was evaluated by assessing the corresponding AUCs.

ROC curve was constructed considering the control population and the Noncompliant GDM mother neonates. The corresponding AUC values of variables investigated were as follows: total renal volume = 0.889, *p* < 0.001; cortical volume = 0.834, *p* < 0.001; *N*-acetyl-β-d-glucosaminidase activity = 0.810, *p* = 0.028; cathepsin B activity = 0.848, *p* = 0.001. Results indicated a good or high diagnostic accuracy.

## 4. Discussion

Fetal exposure to GDM seems to stimulate a negative impact on kidneys [12,13,14,15], and studies demonstrating a direct correlation between reduced nephrogenesis and GDM have indicated this condition as a cause of kidney injury in offspring [12,16]. Low nephron number is considered to be a significant risk factor for kidney disease in later life [8,9,10,11]. This observational study firstly reports data concerning renal development and function in GDM mother newborns at 30–40 days of age. We found that, compared to the control population, kidneys of neonates of GDM mothers who needed insulin therapy and did not reach the goals of treatment [20], which mainly support a strict control of normoglycemia, were characterized by reduced nephrogenesis and tubular impairment and injury. In these, total renal volume and cortical volume, the main surrogate markers of nephron number, were significantly decreased. Additionally, the activities of *N*-acetyl-β-d-glucosaminidase and cathepsin B, indicators of tubular impairment and injury or dysfunction [29,30,31], were significantly increased. Only these two biochemical compounds were modulated by GDM. The β2-microglubulin, a marker of tubule dysfunction [27,28], did not vary in the above populations, possibly indicating that *N*-acetyl-β-d-glucosaminidase and cathepsin B are more specific or earlier indicators of tubule impairment and injury or dysfunction than β2-microglubulin in these neonates, at 30–40 days of postnatal age. Furthermore, such conditions, at this postnatal age, did not seem to be associated with either a general perturbation of lysosomes in tubule or to a kidney dysfunction involving the absorption of macromolecules in renal proximal tubular cells and the remodeling of extracellular matrix proteins in the tubulointerstitial area (events that contribute to the pathogenesis of renal interstitial fibrosis). The urinary activity of the lysosomal enzymes β-glucuronidase and legumain were statistically comparable to the control population. In contrast to data in literature [16], this was also true for urinary levels of albumin, a marker of glomerular function. In 3-year-old GDM children, reduced nephrogenesis was seen to be associated to proteinuria [16]. To date, such a discrepancy is not clear, however, it may be due to a difference in patient age and may indicate that the impairment and injury or dysfunction of the tubule, shown here, represents the very early stage of the tubulointerstitial changes that could progress toward proteinuria and glomerulosclerosis [48].

Interestingly, in the Compliant GDM mother neonates, the above effects of GDM on neonatal renal development and function were not seen. Contrary to the Noncompliant GDM mother subgroup, total renal volume, cortical volume, and *N*-acetyl-β-d-glucosaminidase and cathepsin B activity were similar compared to the control population.

Maternal obesity may have negative effects on neonates [41,42,43,44,45,46,47]. Recognizing three GDM classes in which mother obesity never occurred (Class 1: pre-gestational and gestational BMI < 30), took place only during pregnancy (Class 2: pre-gestational BMI < 30 and gestational BMI > 30), or was present in both pre-gestational and gestational periods (Class 3: pre-gestational and gestational BMI > 30), we found that the negative effects of GDM on kidney development and integrity concerned GDM neonates of mothers who never experienced obesity or experienced it in both pre-gestational and gestational periods. No renal consequences of GDM were seen when gestational obesity was preceded by pre-gestational BMI < 30. In this class, data were similar to controls. Interestingly, such findings emerged only in the Noncompliant GDM mother neonates. As expected, all classes of the Compliant GDM mother’s neonate subgroup were comparable to control. From a first analysis of the above results, we could conclude that the management of GDM mothers, which mainly involved a strict control of normoglycemia, may ensure both normal renal development and integrity in neonates. If maternal GDM is not managed, the kidneys of newborns may be negatively affected when mothers with a pre-conceptional BMI < 30 maintain BMI < 30 during gestation. If BMI comes to be >30 during gestation, a protective effect may occur in the kidney against renal GDM consequences. This does not seem true if mothers are also obese before pregnancy. Thus, when GDM is not managed, the kidneys of newborns may be negatively affected independently of the concurrence of pre-gestational obesity. Gestational obesity alone, not preceded by pre-gestational obesity, may induce a protective condition capable of preventing the adverse renal consequence of GDM. The significance of these results is not yet clear. However, it may be speculated that this trend could be due to a possible lower number, intensity, or duration of the hyperglycemic peaks (whose data, however, were not available retrospectively), or, consistent with data of variability, to a possible important role of gestational weight gain. We found that all classes of the Compliant GDM mother neonate subgroup, and the only class of the Noncompliant GDM mother subgroup, which was unaffected by GDM, exhibited similar maternal weight gain (median: 10–13 kg; mean: 10.5–13.5 kg; interquartile range from a minimum of 8.6 kg to a maximum of 17.3 kg). Hence, a reasonable range of maternal weight gain, such as that found, which could be called “healthy and protective”, may be thought to allow the foetus to prevent the negative renal consequences of GDM and hyperglycemia, independent of the presence or not of gestational obesity, and that from this perspective, such conditions could more easily provide the correct supply of anti-oxidants or other protective nutrients to the foetus [49].

Finally, analysis of the corresponding AUCs indicates that as in the case of IUGR and preterm neonates [32], total renal volume, cortical volume, and the urinary activity of *N*-acetyl-β-d-glucosaminidase and cathepsin B may provide an early indication of GDM neonates at risk of renal disease in later life.

## 5. Conclusions

GDM has recently reached epidemic proportions worldwide and dysregulation of glucose metabolism is found in up to 15% of pregnancies. The importance of glycemic control is crucial, as GDM results in serious negative outcomes at birth for mothers and their offspring, with possible long-term effects on their health [12,13,14,15,16,17,50,51]. As a consequence, these factors must be taken into account and must stimulate the development of preventive intervention strategies, including maternal GDM management and the early identification of GDM neonates at risk of morbidities.

This observational study highlights that GDM impairs both renal development and tubular integrity in neonates at 30–40 days of postnatal age. Such impairment, however, seems to be very early and preventable. An appropriate management of GDM, aiming (as a main goal) at maintaining blood glucose as near to normal as possible, may prevent these negative effects, indicating the fundamental role of strict control of normoglycemia and compliance to a GDM management program [20]. Data also suggest a possible fundamental role of a “healthy and protective” range of weight gain in this condition. Randomized controlled trials should be addressed in this direction in order to validate these observational clinical data.

Hence, in agreement with the DOHaD concept and the recent guidelines [18,19,20,52], prevention and early identification of neonates at risk of hypertension and renal disease in later life due to GDM should involve a multidisciplinary approach, beginning from pre-conceptional maternal counselling and continuing with the early recognition and the follow-up of newborns at risk of disease in the perinatal period. Renal 3D ultrasound technology, which allows measurements of total renal volume and cortical volume, combined with analysis of urinary biomarkers, may represent an improved tool for this purpose. Further studies in order to validate total renal volume, cortical volume, and urinary activity of *N*-acetyl-β-d-glucosaminidase and cathepsin B as early indicators of long-term risk of renal diseases are needed.

## Figures and Tables

**Figure 1 jcm-08-00429-f001:**
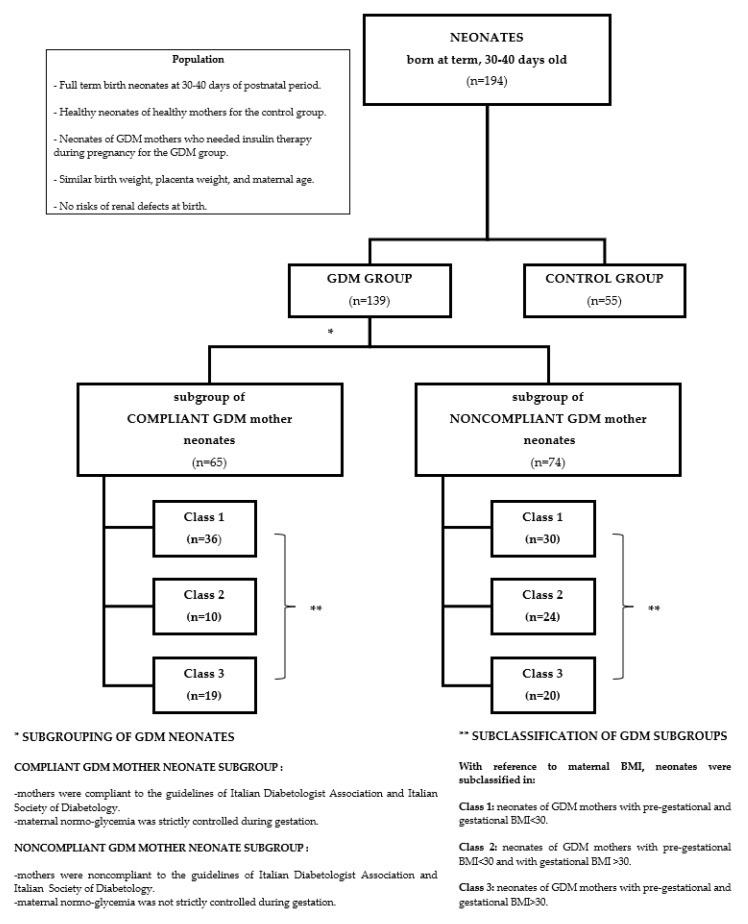
Flow diagram of case selection.

**Figure 2 jcm-08-00429-f002:**
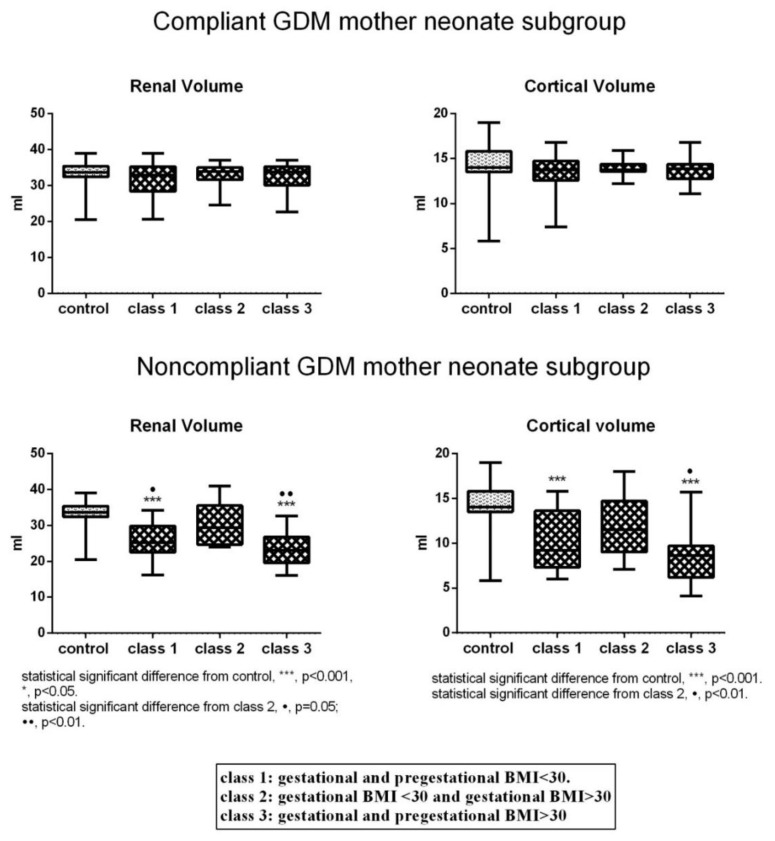
Renal mass parameters in the three classes of Compliant and Noncompliant GDM mother neonate subgroups and control.

**Figure 3 jcm-08-00429-f003:**
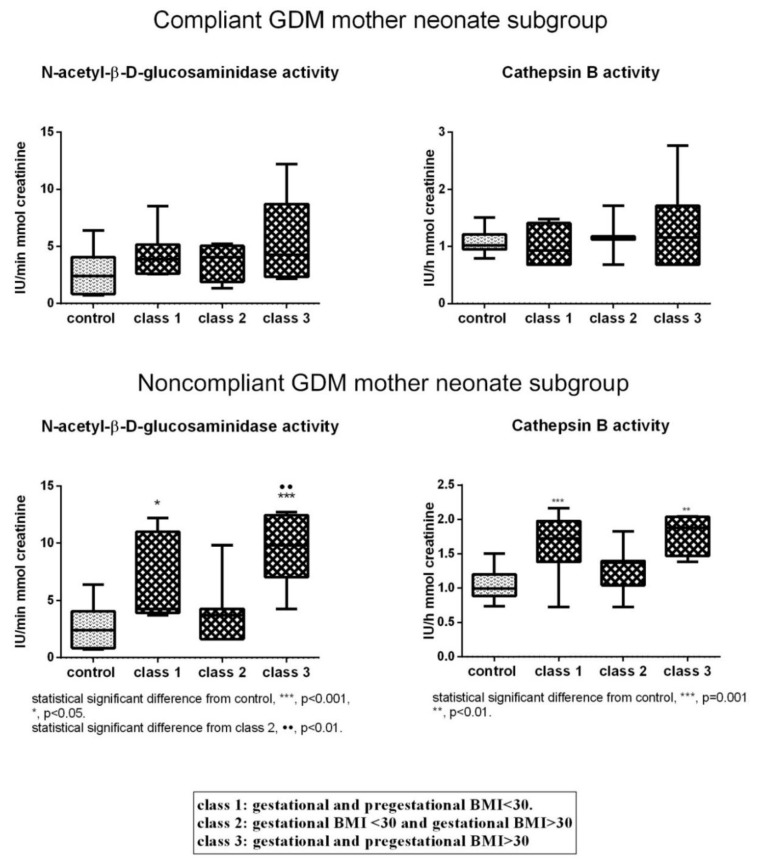
Urinary activity of *N*-acetyl-β-d-glucosaminidase and cathepsin B in the three classes of Compliant and Noncompliant GDM mother neonate subgroups and control.

**Figure 4 jcm-08-00429-f004:**
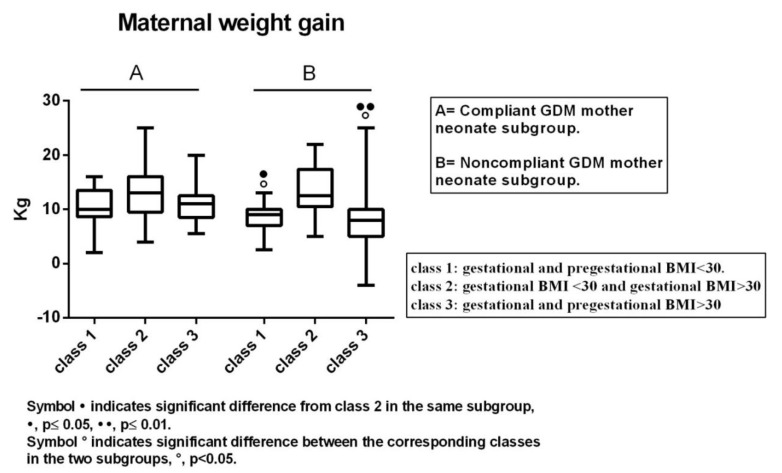
Maternal weight gain in the three classes of Compliant and Noncompliant GDM mother neonates subgroups.

**Table 1 jcm-08-00429-t001:** Characteristics of the study.

**Type**	Observational retrospective
**Aims**	Evaluation of renal development and function in a population of 30–40 days old GDM neonates.
	Evaluation of renal development and function in GDM neonates with reference to different management of GDM during pregnancy.
	Evaluation of the impact of maternal BMI on renal development and function in the GDM population.
**Population**	Whole population, *n* = 194.
	GDM population, *n* = 139.
	Control population, *n* = 55.
**Inclusion Criteria**	Full term birth, characterized by normal physiological postnatal adaptation.
	30–40 days of postnatal period
	Similar birth weight, placenta weight and maternal age
	Apgar score value: ≥7 and ≤10, at the 1st and 5th minute.
	Healthy neonates of healthy mothers, for the control group
	Neonates of GDM mothers who needed insulin therapy during pregnancy, for the GDM group
**Exclusion Criteria**	Prematurity, IUGR, twins, macrosomia, sepsis, asphyxia, any neonatal malformation including those of kidney, AKI.
	Maternal accelerated weight gain in the first trimester, maternal hypertensive disorders, preeclampsia, maternal smoking, maternal alcohol use, maternal caffeine abuse, pre-existent renal diseases in both parents and in family history, maternal diabetes mellitus type I and type II.
**Primary Endpoints**	Assessment of renal volume and cortical volume
	Assessment of urinary albumin, β2-microglubulin, and the activity of *N*-acetyl-β-d-glucosaminidase and cathepsin B
**Secondary Endpoints**	Assessment of the urinary activity of β-glucuronidase and legumain.
	Assessment of the impact of maternal BMI on renal development and function in GDM neonates.
	Assessment of the diagnostic efficiency of renal volume, cortical volume, *N*-acetyl-β-glucosaminidase, and cathepsin B activity as risk factors for long-term renal disease.

GDM: gestational diabetes mellitus, IUGR: intrauterine growth restriction; AKI: acute kidney injury, BMI: body mass index.

**Table 2 jcm-08-00429-t002:** Characteristics of the study population.

	Control	Compliant GDM Mother Neonate Subgroup	Noncompliant GDM Mother Neonate Subgroup
Sex	33 (m), 22 (f)	32 (m), 33 (f)	40 (m), 34 (f)
Birth weight (g)	3308 ± 484.3	3326 ± 407	3318 ± 415
Gestational weeks (week)	39.04 ± 1.9	38.8 ± 1.12	38.9 ± 0.96
Placenta weight (g)	558 ± 92.7	563 ± 84.71	549 ± 89
Maternal age (year)	28.75 ± 4.5	27.5 ± 2.9	29.5 ± 4
Diagnosis of GDM (week)	-	26 ± 1.15	24.1 ± 1.5

Results are expressed as mean ± standard deviation; m: male; f: female.

**Table 3 jcm-08-00429-t003:** BMIs in the GDM mother population.

Subgroup	Class		Pre-Gestational BMI	Gestational BMI
**Compliant**	**Class 1** (*n* = 36)	median	22	26
		IQR	20.8–24	25.2–27.9
		min/max	19–28	21–29
		mean ± sem	22.3 ± 0.3	26.3 ± 0.3
	**Class 2** (*n* = 10)	median	27.5	32
		IQR	26–29	30.8–35
		min/max	24.7–29.6	30–35
		mean ± sem	27.3 ± 0.4	32.2 ± 0.5
	**Class 3** (*n* = 19)	median	32	37
		IQR	30.5–34	34–37.5
		min/max	30–36	33–39
		mean ± sem	32.2 ± 0.7	36.2 ± 0.7
**Noncompliant**	**Class 1** (*n* = 30)	median	24	27
		IQR	22–25	25.25–29
		min/max	15–27	18–29.9
		mean ± sem	23.25 ± 0.4	26.5 ± 0.4
	**Class 2** (*n* = 24)	median	27	32
		IQR	25.7–28	31–33
		min/max	24.6–29	30.1–33.8
		mean ± sem	27 ± 0.4	32 ± 0.4
	**Class 3** (*n* = 20)	median	33	36.3
		IQR	31.2–35.25	34.8–38.4
		min/max	30–45	31–46
		mean ± sem	34 ± 0.7	37 ± 0.8

IQR: interquartile range; sem: standard error of mean.

**Table 4 jcm-08-00429-t004:** Renal mass parameters and urinary biomarkers in neonates of GDM group, Compliant, and Noncompliant GDM mother subgroups and control.

**a. Renal Mass Parameters.**					
		**Control** **(*n* = 55)**	**GDM Group (*n* = 139)**	**Compliant GDM Mother Subgroup (*n* = 65)**	**Noncompliant GDM Mother Subgroup (*n* = 74)**
Renal volume (mL)	median	33.7	29.2 ***	33.4	24.8 ***^,^ °°°
	IQR	32.18–35.48	24.8–33.6	28.75–35	22.15–29.5
	min/max	27.8–39	16.1–41	20.6–39	16.1–41
	mean ± sem	33.69–0.33	29 ± 0.49	32 ± 0.55	25.6 ± 0.75
Cortical volume (mL)	median	14.00	12.8 ***	13.8	9.2 ***^,^ °°°
	IQR	13.5–15.8	9.6–14	12.95–14.53	7.2–13.00
	min/max	5.8–19	4.1–18	7.4–16.8	4.1–18
	mean ± sem	14.4 ± 0.27	11.95 ± 0.27	13.67 ± 0.22	10.1 ± 0.47
**b. Urinary Biomarkers**					
		**Control** **(*n* = 55)**	**GDM Group (*n* = 139)**	**Compliant GDM Mother Subgroup (*n* = 65)**	**Noncompliant GDM Mother Subgroup (*n* = 74)**
Albumin (mg/mmol creatinine)	median	5.9	9.06	7.84	9.3
	IQR	4.26–9.6	4.3–9.8	3.5–10.94	7.6–9.57
	min/max	2.2–20.5	2.7–18.5	2.72–18.49	6.96–10.33
	mean ± sem	7.45 ± 1.11	8.17 ± 0.79	7.99 ± 10.2	8.8 ± 0.5
β2microglobulin (μg/mmol creatinine)	median	0.35	0.41	0.40	0.44
	IQR	0.17–0.67	0.30–1	0.38–3.45	0.18–0.96
	min/max	0.5–1.05	0.05–3.45	0.38–3.45	0.05–0.96
	mean ± sem	0.44 ± 0.14	0.83 ± 0.35	1.41 ± 1.02	0.53 ± 0.17
Cathepsin B (IU/h mmol creatinine)	median	0.99	1.41 *	1.18	1.43 *
	IQR	0.88–1.2	1.04–1.86	0.68–1.7	1.37–1.98
	min/max	0.73–1.51	0.68–1.86	0.68–2.77	0.73–2.2
	mean ± sem	1.04 ± 0.05	1.47 ± 0.13	1.38 ± 0.23	1.65–0.12
*N*-acetyl-β-d-glucosaminidase (IU/min mmol creatinine)	median	2.38	4.12 *	3.95	4.28 *
	IQR	0.83–4.05	2.56–6.87	2.43–5.08	3.71–12.2
	min/max	0.71–6.39	1.36–12.71	1.36–12.19	1.63–12.71
	mean ± sem	2.66 ± 0.52	5.29 ± 0.78	4.47 ± 0.76	6.92 ± 1.71

Symbol * indicates significant difference from Control; * *p* < 0.05, *** *p* < 0.001; symbol ° indicates significant difference from Compliant GDM mother neonate subgroup; °°° *p* < 0.001.

**Table 5 jcm-08-00429-t005:** Urinary activity of β-glucuronidase and legumain in neonates of Compliant and Noncompliant GDM mother subgroups and control.

		Control (*n* = 55)	Compliant GDM Mother Neonate Subgroup (*n* = 65)	Noncompliant GDM Mother Neonate Subgroup (*n* = 74)
**β-glucuronidase** (IU/min mmol creatinine)	median	0.97	0.98	1.12
	IQR	0.45–1.7	0.71–1.6	0.78–1.65
	min/max	0.3–2.72	0.02–7.5	0.57–2.15
	mean ± sem	1.15 ± 1.19	1.38 ± 0.2	1.22 ± 0.13
**legumain** (IU/h mmol creatinine)	median	1.77	0.18	0.186
	IQR	0.15–0.27	0.1–0.27	0.13–0.5
	min/max	0.12–0.54	0.016–0.48	0.11–0.58
	mean ± sem	0.23 ± 0.05	0.2 ± 0.04	1.12 ± 0.13

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
