# Peer review of "Renal Consequences of Gestational Diabetes Mellitus in Term Neonates: A Multidisciplinary Approach to the DOHaD Perspective in the Prevention and Early Recognition of Neonates of GDM Mothers at Risk of Hypertension and Chronic Renal Diseases in Later Life"

_jcm, 2019, doi:10.3390/jcm8040429_

Reviewer 1 Report

This is an interesting paper correlating urinary and kidney function parameters in newborns (at 30-40 days of life) with the glycemic control of their mothers, affected by gestational diabetes on insulin treatment.

From the obstetrical point of view, the study is well designed and correctly conducted.

I would suggest however an in depth nephrologist evaluation of urinary tests in newborns.

Author Response

Reviewer 1.

Comments and Suggestions for Authors

 This is an interesting paper correlating urinary and kidney function parameters in newborns (at 30-40 days of life) with the glycemic control of their mothers, affected by gestational diabetes on insulin treatment.

From the obstetrical point of view, the study is well designed and correctly conducted.

 Point 1:  I would suggest however an in depth nephrologist evaluation of urinary tests in newborns.

Response 1: We suppose that this observation is addressed to the Editors. However, we wish to reassure Reviewer 1 that, except for legumain, all other urinary biochemical parameters examined have previously and extensively been studied in newborns in order to evaluate renal function and maturation.

Aisa, M.C.; Cappuccini, B.; Barbati, A.; Orlacchio, A.; Baglioni, M.; Di Renzo, G.C. Biochemical parameters of renal impairment /injury and surrogate markers of nephron numbers in intrauterine growth restricted and preterm neonates at 30-40 days of postnatal corrected age  Pediatr. Nephrol. 2016, 31,  2277-2287.

Askenazi DJ, Koralkar R, Patil N, Halloran B, Ambalavanan N, Griffin R. Acute kidney injury urine biomarkers in very low-birth-weight infants. Clin J Am Soc Nephrol. 2016;11(9):1527-35.

Awad H, el-Safty I, el-Barbary M, Imam S (2002) Evaluation of renal glomerular and tubular functional and structural integrity in neonates. Am J Med Sci 324:261-266.

Chen JY, Lee YL, Liu CB (1991) Urinary beta 2-microglobulin and N-acetylbeta- D-glucosaminidase (NAG) as early markers of renal tubular dysfunction in sick neonates. J Formos Med Assoc 90:132-137.

Clark PM, Bryant TN, Hall MA, Lowes JA, Rowe DJ (1989) Neonatal renal function assessment. Arch Dis Child 64:1264-1269.

DeFreitas MJ, Seeherunvong W, Katsoufis CP, RamachandraRao S, Duara S, Yasin S, Zilleruelo G, Rodriguez MM, Abitbol CL. Longitudinal patterns of urine biomarkers in infants across gestational ages. Pediatr Nephrol. 2016;31(7):1179-88.

Fell JM, Thakkar H, Newman DJ, Price CP (1997) Measurement of albumin and low molecular weight proteins in the urine of newborn infants using a cotton wool ball collection method. Acta Paediatr 86:518-522.

Galaske RG (1986) Renal functional maturation: renal handling of proteins by mature and immature newborns. Eur J Pediatr 145:368-371.

Gubhaju L, Sutherland MR, Horne RS, Medhurst A, Kent AL, Ramsden A, Moore L, Singh G, Hoy WE, Black MJ (2014) Assessment of renal functional maturation and injury in preterm neonates during the first month of life. Am J Physiol Renal Physiol 307:149-158.

Hayashi M, Tomobe K, Hirabayashi H, Hoshimoto K, Ohkura T, Inaba N (2001) Increased excretion of N-acetyl-beta-D-glucosaminidase and beta-2-microglobulin in gestational week . Am J Med Sci 321:168-172.

Heimann G. Renal toxicity of aminoglycosides in the neonatal period. Pediatr Pharmacol (New York). 1983;3:251-7.

HodovanetsJ, Babintseva A, Longford NT, Agafonova L. Urinary protein markers of renal dysfunction in full-term newborns with disorders of early neonatal period.  Journal of Pediatric and Neonatal Individualized Medicine 2018;7(1):e07010

Kojima T, Sasai-Takedatsu M, Hirata Y, Kobayashi Y (1994) Characterization of renal tubular damage in preterm infants with renal failure. Acta  Paediatr Jpn 36:392–395.

Perrone S, Mussap M, Longini M, Fanos V, Bellieni CV, Proietti F, Cataldi L, Buonocore G (2007) Oxidative kidney damage in preterm newborns during perinatal period. Clin Biochem 40:656-660.

Tsukahara H, Fujii Y, Tsuchida S, Hiraoka M, Morikawa K, Haruki M, Sudo M (1994) Renal handling of albumin and beta-2-microglobulin in neonates. Nephron 68:212-216.

Tsukahara H, Hori C, Tsuchida S, Hiraoka M, Sudo M, Haruki S, Suehiro F (1994) Urinary N-acetyl-beta-D-glucosaminidase excretion in term and preterm neonates. J Paediatr Child Health 30:536-538.

Tsukahara H, Yoshimoto M, Saito M, Sakaguchi T, Mitsuyoshi I, Hayashi S, Nakamura K, Kikuchi K, Sudo M (1990) Assessment of tubular function in neonates using urinary beta 2-microglobulin. Pediatr Nephrol 4:512-514.

 Reviewer 2 Report

 Renal Consequences of Gestational Diabetes Mellitus in Term Neonates: A Multidisciplinary Approach on Dohad Perspective in the Prevention and Early Recognition of Neonates of GDM Mothers at Risk of Hypertension and Chronic Renal Diseases in Later Life

Renal volumes and urinary biomarkers of renal function and tubular impairment/injury were evaluated in 30-40 days old newborns of GDM mothers who needed insulin therapy during pregnancy. Neonates of mothers who maintained a strict control of normoglycemia during pregnancy and fulfilled the other criteria of GDM management program showed no differences compared to controls. Conversely, those of mothers who did not maintain the glycemic control, were not compliant to the management program exhibited significantly lower levels of renal volumes and higher activity of N-acetyl--D-glucosaminidase and cathepsin B. Differences on maternal pregestational and gestational body mass index, as well as on maternal weight gain were also demonstrated.

The findings indicate that a multidisciplinary approach which involves an appropriate management of GDM prevents the negative effects of GDM on kidney at 30-40 days of  postnatal age, indicating a fundamental role of the glycemic control as well as of an adequate range of maternal weight gain.

Total renal volume, cortical volume and urinary N-acetyl--D glucosaminidase and cathepsin B activity may be suggested as indicators for the early recognition of GDM neonates at long-term risk of hypertension and kidney disease.

The article is of great interest and I congratulate authors for this publication!

I am sure that this study will be cited a lot.

However, I have several minor comments to the authors:

Please add the sample size to the abstract.

Introduction: The association between GDM and long-term consequences to the offspring was indeed previously investigated. Authors should cite for an example a very recent publication pointing out cardiovascular complications, including hypertensive disorders to offspring of diabetic mothers (Leybovitz-Haleluya N, Acta Diabetol. 2018 Jun 23. doi: 10.1007/s00592-018-1176-1. [Epub ahead of print] Maternal gestational diabetes mellitus and the risk of subsequent pediatric cardiovascular diseases of the offspring: a population-based cohort study with up to 18 years of follow up).

Results: Please add data regarding maternal hypertensive disorders.

Since GDM and preeclampsia are significantly associated, it would be important to control for preeclampsia.

Please add data on preterm delivery (Table 1).

Can authors give data on Apgar scores?

Does mode of delivery have any influence on the outcome? Please give data on CS rate.

Can authors give data on BMI of the parturients?

Please add data regarding parity.

Discussion: The importance of glycemic control is crucial! Interestingly even in the mothers, high glucose level during pregnancy, even if within the range of slight glucose intolerance, may serve as a marker for future maternal atherosclerotic morbidity (Diabet Med. 2016 Jul;33(7):920-5. Can slight glucose intolerance during pregnancy predict future maternal atherosclerotic morbidity?).

Author Response

Reviewer 2

Comments and Suggestions for Authors

 Renal Consequences of Gestational Diabetes Mellitus in Term Neonates: A Multidisciplinary Approach on Dohad Perspective in the Prevention and Early Recognition of Neonates of GDM Mothers at Risk of Hypertension and Chronic Renal Diseases in Later Life

Renal volumes and urinary biomarkers of renal function and tubular impairment/injury were evaluated in 30-40 days old newborns of GDM mothers who needed insulin therapy during pregnancy. Neonates of mothers who maintained a strict control of normoglycemia during pregnancy and fulfilled the other criteria of GDM management program showed no differences compared to controls. Conversely, those of mothers who did not maintain the glycemic control, were not compliant to the management program exhibited significantly lower levels of renal volumes and higher activity of N-acetyl-b-D-glucosaminidase and cathepsin B. Differences on maternal pregestational and gestational body mass index, as well as on maternal weight gain were also demonstrated.

The findings indicate that a multidisciplinary approach which involves an appropriate management of GDM prevents the negative effects of GDM on kidney at 30-40 days of  postnatal age, indicating a fundamental role of the glycemic control as well as of an adequate range of maternal weight gain.

Total renal volume, cortical volume and urinary N-acetyl-b-D glucosaminidase and cathepsin B activity may be suggested as indicators for the early recognition of GDM neonates at long-term risk of hypertension and kidney disease.

The article is of great interest and I congratulate authors for this publication!

I am sure that this study will be cited a lot.

We really thank Reviewer 2 for his congratulations and his precious support for the constructive feedback aimed at improving our manuscript.

 However, I have several minor comments to the authors:

Point 1:  Please add the sample size to the abstract.

Response 1: We have corrected the abstract as indicated.

Point 2:. Introduction: The association between GDM and long-term consequences to the offspring was indeed previously investigated. Authors should cite for an example a very recent publication pointing out cardiovascular complications, including hypertensive disorders to offspring of diabetic mothers (Leybovitz-Haleluya N, Acta Diabetol. 2018 Jun 23. doi: 10.1007/s00592-018-1176-1. [Epub ahead of print] Maternal gestational diabetes mellitus and the risk of subsequent pediatric cardiovascular diseases of the offspring: a population-based cohort study with up to 18 years of follow up).

Response 2: We would like to thank Reviewer 2 for this suggestion and, accordingly, we have improved the introduction.

Point 3a:  Results: Please add data regarding maternal hypertensive disorders.

Point 3b: Since GDM and preeclampsia are significantly associated, it would be important to control for preeclampsia.

-Point 3c : Please add data on preterm delivery (Table 1).

Response 3a-3b-3c: : Prematurity, maternal hypertensive disorders, preeclampsia were exclusion criteria. To make clearer these aspects, we have modified the text and added a new table (Table 1. Characteristics of the study).

Point 4:  Can authors give data on Apgar scores?

Response 4: All the enrolled neonates showed an Apgar score value ≥7 and ≤10, at the 1st and 5th minute. We have added these data in the inclusion criteria.

Point 5: Does mode of delivery have any influence on the outcome? Please give data on CS rate.

Response 5: We did not investigate the influence of delivery. We found a Cs rate of nearly 42.5 %

Point 6: Can authors give data on BMI of the parturients?

Response 6: Reviewer 2 is right. Accordingly, we have included data of maternal BMIs in a new table (Table 3).

Point 7 : Please add data regarding parity.

Response 7: We did not consider this variable and its possible influence. This indication stimulates our future investigations.

Point 8: Discussion: The importance of glycemic control is crucial! Interestingly even in the mothers, high glucose level during pregnancy, even if within the range of slight glucose intolerance, may serve as a marker for future maternal atherosclerotic morbidity (Diabet Med. 2016 Jul;33(7):920-5. Can slight glucose intolerance during pregnancy predict future maternal atherosclerotic morbidity?).

Response 8: Reviewer 2 highlights the significance of the glycemic control during pregnancy. It is right. It is important to emphasize it. We would like to thank him for this additional indication. Accordingly, we improved our Conclusions.

 The English revision has been done.

Reviewer 3 Report

This manuscript reports an observational study of the impact of gestational diabetes mellitus on kidney structure and some markers of kidney function in the offspring.

The topic addressed is clinically relevant, as GDM is currently considered a possible cause of the worldwide spreading of type 2 diabetes, and exposes the offspring to an increased risk of chronic disease at adulthood, including altered renal function.

This studies suggests that GDM, when poorly controlled, is responsible for a decreased renal volume and altered urinary markers of renal function in the offspring.

However, some important issues in the design of this observational study may induce bias in the results and their interpretation, and need to be clarified:

the study design, prospective or retrospective? Is it a chart review, or were the patients examined within the purpose of the study ?

were criteria for classification of the patients (mothers and infants) pre-defined or not? They must be described.

the definition of compliant and non compliant mothers is qualitative and objective criteria are needed.

seemingly, an error affects the patients flow chart (the total of patients in the second raw is higher than the total of the patients enrolled in the study).

did the mothers give their consent to the study, is an ethical review board approval available?

the data of this observational study are discussed and described with terms that suggest causality. It must be clear that the study methods only allow the generation of hypotheses, and any wording hinting at a causal association should be avoided. The conclusion should be more cautious.   

 Author Response

Reviewer 3

Comments and Suggestions for Authors

 This manuscript reports an observational study of the impact of gestational diabetes mellitus on kidney structure and some markers of kidney function in the offspring.

The topic addressed is clinically relevant, as GDM is currently considered a possible cause of the worldwide spreading of type 2 diabetes, and exposes the offspring to an increased risk of chronic disease at adulthood, including altered renal function.

This studies suggests that GDM, when poorly controlled, is responsible for a decreased renal volume and altered urinary markers of renal function in the offspring.

However, some important issues in the design of this observational study may induce bias in the results and their interpretation, and need to be clarified:

 Point 1: the study design, prospective or retrospective? Is it a chart review, or were the patients examined within the purpose of the study ?

Response 1: We would like to thank Reviewer 3 for these comments. We agreed and, accordingly, we have changed our manuscript including a table (Table 1) which resumes the characteristics of the study.

Point 2. were criteria for classification of the patients (mothers and infants) pre-defined or not? They must be described.

Response 2: The classification in Compliant and Noncompliant mothers was already predefined. Conversely, the categorization of GDM mothers in the three subclasses (based on the BMIs) was taken into account during the course of the present study, in order to evaluate the impact of maternal BMI on kidney development and function in the population of GDM neonates. We further detailed these aspects in the sections of GDM mothers and neonates in Materials and Methods.

 Point 3: the definition of compliant and noncompliant mothers is qualitative and objective criteria are needed.

Response 3: Thanks to Reviewer 3 for his recommendation. He is right. We have included the objective criteria of Compliant and Noncompliant mothers definition in the text.

Point 4: seemingly, an error affects the patients flow chart (the total of patients in the second raw is higher than the total of the patients enrolled in the study).

Response 4: Thanks to Reviewer 3 for this advice. We have corrected the error.

Point 5: did the mothers give their consent to the study, is an ethical review board approval available?

Response 5: An Institutional review board approval was obtained for data collection and mothers were informed and gave a specific consent to the study (and helped the Authors in the phase of urine collection).

Point 6: the data of this observational study are discussed and described with terms that suggest causality. It must be clear that the study methods only allow the generation of hypotheses, and any wording hinting at a causal association should be avoided. The conclusion should be more cautious.

Response 6: Reviewer 3 is right. Accordingly, we have modified our conclusions.

 The English revision has been done.

Round  2

Reviewer 3 Report

The Authors have relied appropriately to all my remarks.